# *Bacillus subtilis* Edible Films for Strawberry Preservation: Antifungal Efficacy and Quality at Varied Temperatures

**DOI:** 10.3390/foods13070980

**Published:** 2024-03-22

**Authors:** Jesús Rubén Torres-García, Arnulfo Leonardo-Elias, María Valentina Angoa-Pérez, Edgar Villar-Luna, Sergio Arias-Martínez, Guadalupe Oyoque-Salcedo, Ernesto Oregel-Zamudio

**Affiliations:** 1Instituto Politécnico Nacional, Centro Interdisciplinario de Investigación para el Desarrollo Integral Regional (CIIDIR), Unidad Michoacán, Justo Sierra 28, Col. Centro, Jiquilpan 59510, Michoacán, Mexico; jrtorresg@ipn.mx (J.R.T.-G.); aelias1700@alumno.ipn.mx (A.L.-E.); vangoa@ipn.mx (M.V.A.-P.); evillarl@ipn.mx (E.V.-L.); sariasm@ipn.mx (S.A.-M.); 2Investigadores por México, Consejo Nacional de Humanidades, Ciencias y Tecnología (CONAHCYT), México City 03940, Mexico; 3Tecnológico Nacional de México, Instituto Tecnológico de Roque, Carretera Celaya—Juventino Rosas Km. 8, Celaya 38110, Guanajuato, Mexico

**Keywords:** *Bacillus subtilis*, edible films, strawberry preservation, *Rhizopus stolonifer*, antifungal properties

## Abstract

Fungal infestations, particularly from *Rhizopus stolonifer*, pose significant post-harvest challenges for strawberries, compromising their shelf life and quality. Traditional preservation methods, including refrigeration, offer limited protection against such pathogens. This study introduces an innovative approach, utilizing edible films infused with *Bacillus subtilis* strains GOS 01 B-67748 and HFC 103, known for their antifungal properties. We demonstrate that these bioactive films not only inhibit fungal growth effectively but also enhance the preservation of strawberries at varying temperatures. The inclusion of *Bacillus subtilis* in edible films represents a significant advancement in extending the viability of strawberries, surpassing the efficacy of conventional methods. Our findings suggest a promising avenue for natural, safe food preservation techniques, aligning with current consumer preferences for additive-free products. This research contributes to the broader understanding of microbial-based food preservation strategies, offering potential applications across a range of perishable commodities.

## 1. Introduction

Strawberries are particularly susceptible to fungal infestations, largely due to their high moisture content and tendency to soften quickly. The most common post-harvest fungal pathogens affecting strawberries are *Botrytis cinerea* and *Rhizopus stolonifer* [1,2]. While refrigeration (1–4 °C) is widely used to prolong their shelf life, it is not entirely effective against decay caused by fungi such as *Aspergillus*, *Penicillium*, *Alternaria*, *Cladosporium*, *Botrytis*, and *Rhizopus*. These fungi can still proliferate, potentially leading to losses of up to 50% [3,4,5]. To combat this, edible films or coatings have been proposed as innovative solutions to extend the freshness of strawberries.

Various formulations of films incorporating antifungal agents have shown promise in enhancing the shelf life of strawberries under different storage conditions. For example, a film made from pectin and eugenol (1% and 0.2%, respectively) significantly inhibited fungal and yeast growth for up to 13 and 14 days at a storage temperature of 0.5 °C, surpassing the performance of the control group without any film [6]. Likewise, strawberries coated with a chitosan, citric acid, and glycerin film (1.5%) exhibited an extended shelf life of 14 days at approximately 5 °C, effectively curtailing microbial growth [7]. Furthermore, a coating comprising gelatin and *Mentha pulegium* (4% and 1%, respectively) extended the longevity of strawberries to 13 days at 4 °C, with treated fruits showing 60% less deterioration and a significant reduction in yeast and mold growth compared to untreated counterparts [8].

The integration of microorganisms into film formulations to bolster defense against phytopathogens has also been explored, with lactic acid-producing bacteria such as *Lactobacillus plantarum* receiving notable attention. The inclusion of *Lactobacillus plantarum* in a potato starch film effectively countered *Botrytis cinerea* on grapes, achieving a complete reduction in incidence and severity after 7 days at 20 °C [9]. The potential of other microorganisms, including *Bacillus subtilis*, for fungal control, though less investigated, has shown promise. *Bacillus subtilis* has demonstrated effectiveness in inhibiting *R. stolonifer* growth, particularly through the secretion of antimycotic lipopeptides that disrupt fungal membrane permeability or induce cell death through other mechanisms [10].

Incorporating bacteria into edible film formulations could further enhance the control of phytopathogens in fruit preservation efforts. A film containing cyclolipopeptides from *Bacillus subtilis* significantly reduced fungal populations on blueberries and effectively maintained their freshness [11]. Given the GRAS (Generally Recognized As Safe) status of *Bacillus subtilis*, its application in food preservation is deemed safe for human consumption [12]. This safety is supported by studies demonstrating no adverse effects from high-dose *Bacillus subtilis* ingestion and genomic analyses confirming the absence of antibiotic resistance or virulence genes [13,14,15]. Building on prior research that showed promising results using *Bacillus subtilis*-treated strawberries with a guar gum-based film [16], our study aims to evaluate the effectiveness of a novel edible film formulation enriched with *Bacillus subtilis* strains in enhancing the preservation of strawberries. We focus on determining the antifungal efficacy against *Rhizopus stolonifer* and assessing the impact on the quality of strawberries stored under various temperature conditions.

## 2. Materials and Methods

### 2.1. Biological Materials and Reagents

We utilized two strains of *Bacillus subtilis*, GOS 01 B-67748, registered with the Northern Regional Research Laboratory, and HFC 103, selected for their effectiveness in inhibiting *Rhizopus stolonifer* in vitro. The GOS 01 B-67748 strain was isolated from the rhizosphere of *Solanum lycopersicum* cv. Cerasiforme in a temperate climate, while HFC 103 was derived from the rhizosphere of *Fragaria vesca* in a colder region. *R. stolonifer*, the target pathogen, was isolated from strawberries of the Fortuna variety showing symptoms of infection. Both microorganisms are preserved in the collection of the Phytopathology Laboratory at the Instituto Politécnico Nacional, CIIDIR Unit, in Michoacán, Mexico, where they are maintained under optimal conditions to ensure their viability and pathogenicity for use in this research. We selected strawberries from a crop that had not been previously inoculated with any microorganisms, including *B. subtilis*. Although strawberries may naturally harbor their own native microbiota, our prior analyses confirmed the absence of Bacillus subtilis in these samples. The strawberries used in the experiments, *Fragaria x ananassa* Duch. cv. Cabrillo, were carefully selected based on uniformity in size, shape, weight, and color, and were examined to ensure they were free from any physical damage or visible signs of fungal contamination. These strawberries were harvested from the renowned strawberry-producing region of La Luz in the Pajacuarán municipality, Michoacán de Ocampo, Mexico, known for its high-quality fruit. For the film composition, we used guar gum (Sigma Aldrich, Deisenhofen, Germany), glycerol with a purity of ≥99% (Sigma Aldrich, Deisenhofen, Germany), and food-grade refined candelilla wax sourced from Ceras Naturales Mexicanas, S.A. de C.V., Saltillo, Coahuila, Mexico. These components were selected for their purity, consistency, and compliance with food safety standards, ensuring that the resulting edible films would be safe for application on strawberries intended for consumption. Guar gum was chosen for its superior film-forming properties, glycerol was used as a plasticizer to enhance film flexibility, and refined candelilla wax was incorporated for its moisture barrier capabilities, contributing to the effectiveness and safety of the edible films in extending the shelf life of strawberries.

### 2.2. Bacillus subtilis Suspension Preparation

To prepare the *B. subtilis* suspensions, we initially cultured each strain on potato dextrose agar (PDA) plates and incubated them at 37 °C for 24 h. The pH of the suspension was 6.8 ± 0.1. Following incubation, we harvested the bacterial colonies and resuspended them in 100 mL of sterile distilled water within a 500 mL Erlenmeyer flask. We then determined the bacterial concentration using the most probable number (MPN) method, correlating it with optical density measurements at 520 nm to generate a calibration curve. We achieved a target concentration of 1 × 10^12^ CFU/mL, as indicated by an optical density of 0.99 absorbance, measured using a UV-visible spectrophotometer (Perkin Elmer, model Lambda 2, serial number 5109).

### 2.3. Edible Film Development

To prepare the films, we adapted the method described by Oregel et al. (2017) [17], with minor adjustments. Our film formulation consisted of 0.8% guar gum, 0.4% candelilla wax, 0.2% glycerol, and a 20% bacterial suspension of *B. subtilis*, using either the GOS 01 B-67748 or HFC 103 strain at a density of 1 × 10^12^ CFU/mL, respectively. The process began by dissolving candelilla wax in distilled water at 80 °C in a sterilized 1000 mL blender jar. Subsequently, guar gum and glycerol were added, and the mixture was homogenized at high speed for 5 min. For the bacterial films, the *B. subtilis* suspension was added to the cooled emulsion (at 35 °C) and manually stirred for 2 min. This resulted in three types of films: a base edible film (Film), a film with *B. subtilis* GOS 01 B-67748 (Film_BsG), and a film with *B. subtilis* HFC 103 (Film_BsH). Figure 1 illustrates an example of the application of edible films with *B. subtilis*, showcasing strawberries coated with these edible films, as well as preformed films.

### 2.4. Physicochemical Properties of Films

Pre-formed films were created by dispensing 20 mL of each film emulsion into sterile Petri dishes (90 × 15 mm) and drying them on a flat surface. Initially, the films underwent a 2 h drying phase under a laminar flow hood (NuAire UN-201-43, series 63) at an ambient temperature of 25 °C ± 5, followed by a further 24 h drying period in an oven (INA-100 DP, Craft, USA) set at 30 °C ± 1. After the drying process, the films were removed with sterile gloves and sealed in Ziploc bags (16.5 cm × 14.9 cm) for later analysis. The film thickness was measured at ten randomly chosen points on each film using a digital caliper (215140, OBI, USA). For density measurements, 2 cm^2^ pieces of film were cut, and their dimensions (width, length, and thickness) were recorded to calculate their volume. Each piece was then weighed on an analytical balance (ADAM, model ABC plus 600H) to determine its mass, and this procedure was performed in triplicate for precision. The density of each film was calculated by dividing its mass by its volume. Moisture content was evaluated by placing 2 cm^2^ film samples on aluminum foil and exposing them to 100% relative humidity at 25 °C for 24 h in a climate chamber. Subsequently, the samples were dried in a natural convection oven at 105 °C for another 24 h. After drying, the samples were weighed on a thermobalance (MS 70, COBOS, USA) for 10 min to ascertain the weight difference before and after drying, facilitating the calculation of moisture percentage. This assessment was replicated three times to ensure result consistency. To assess solubility, moisture-free 2 cm^2^ pieces of the pre-formed film were submerged in 80 mL of distilled water in a 250-mL Erlenmeyer flask. A 3.5 cm-long hexagonal magnetic stir bar was added, and the flask was placed on a magnetic stirrer (Hanna HI 303N-1, USA) set at 1000 rpm for 10 min. Post stirring, the remaining film residue was collected on Whatman #40 filter paper and dried at 60 °C for 24 h to a constant weight. The solubility percentage was then calculated based on the weight difference observed before and after the solubility test.

### 2.5. Rizopus stolonifer Challenge Tests In Vitro

In the in vitro challenge, we assessed the antagonistic effect of *B. subtilis* strains against *R. stolonifer* using PDA agar in Petri dishes (90 × 15 mm). A 5 mm diameter piece of agar heavily colonized by *R. stolonifer* was positioned near one edge of each dish, about 0.2 cm from the perimeter. From the center to the opposite edge of the dish, a loopful of *B. subtilis* was streaked across the agar surface. For controls, dishes were inoculated solely with the fungus. Following incubation at 25 °C, fungal growth was measured at three designated points, namely, the center, left side, and right side, originating from the center of the initial fungal inoculum, with the entire assay conducted in triplicate. To quantify the inhibitory effect of *B. subtilis*, we calculated the percentage of inhibition based on the formula “% Inhibition = ((R1 − R2)/R1) × 100”, where “R1” represents the average fungal growth observed on control plates, and “R2” denotes the average growth on plates treated with *B. subtilis.* This calculation, performed in triplicate for accuracy, follows the method outlined by Singh (2002) [17].

### 2.6. Evaluation of Bacillus subtilis Films against Rhizopus stolonifer

#### 2.6.1. In Vitro Antifungal Efficacy

We conducted challenges with *R. stolonifer* on strawberries to evaluate the protective efficacy of various edible film treatments. Strawberries were coated with the films by dipping them for about 2 s, followed by drying under a laminar flow hood on a hexagonal mesh for 1 h. Subsequently, the film-coated strawberries were transferred to sterile plastic containers (20 × 20 × 6 cm) lined with sterile paper towels. Under sterile conditions, the strawberries were lightly misted with sterile drinking water and inoculated with *R. stolonifer* spores at a concentration of 1 × 10^6^ spores/mL. The containers were then covered with organza cloth to prevent fruit fly intrusion, and the lids were placed loosely on top. The strawberries were stored at 25 °C for a period of 5 days. The study included nine treatments, each with eight strawberries per container and three replicates: untreated strawberries (Swt), strawberries inoculated with *R. stolonifer* (Stw_Rs), strawberries coated with the base film and inoculated with *R. stolonifer* (Film_Rs), strawberries treated with *B. subtilis* HFC 103 and inoculated with *R. stolonifer* (BsH_Rs), strawberries treated with *B. subtilis* GOS 01 B-67748 and inoculated with *R. stolonifer* (BsG_Rs), strawberries coated with an edible film containing HFC103 and inoculated with *R. stolonifer* (Film_BsH_Rs), and strawberries coated with a film containing GOS 01 B-67748 and inoculated with *R. stolonifer* (Film_BsG_Rs). The experimental setup was completely randomized, and the severity of fungal damage was assessed over five days using a 5-point severity scale as proposed by Oregel et al. (2017) [16], where 1 represents 0–24% damage, 2 is for 25–49%, 3 for 50–74%, 4 for 75–99%, and 5 for 100% damage. In the scale, level 1 indicates strawberries that are firm and without visible alterations, suggesting minimal or no impact from the pathogen. Level 2 is characterized by the presence of spots and slight softening of the texture, indicating the onset of infection. At level 3, the emergence of mycelium is observed, signaling an active fungal infection and further deterioration. Level 4 is marked by extensive mycelium development accompanied by a significant softening of the fruit, fluid secretion, and the presence of foul odor, reflecting an advanced state of decomposition. Finally, level 5 represents the total dehydration of the fruit, indicating a complete loss of its structure and quality, as a result of severe and prolonged fungal infection.

#### 2.6.2. Protective Efficacy on Strawberries

In this preservation assay, we examined the impact of six treatments on strawberry damage due to native fungi, using a completely randomized experimental design. Following the application of films as previously described, the strawberries were stored in sterile plastic containers (20 × 20 × 6 cm). The treatments were assessed at two storage conditions: 25 °C for 20 days and 4 °C for 30 days, with each treatment replicated 24 times at each temperature. The six treatments included untreated strawberries (Stw), strawberries with the base film (Film), strawberries treated with *B. subtilis* HFC 103 (BsH), strawberries treated with *B. subtilis* GOS 01 B-67748 (BsG), strawberries coated with a film containing *B. subtilis* HFC103 (Film_BsH), and strawberries coated with a film containing *B. subtilis* GOS 01 B-67748 (Film_BsG). Damage severity was assessed using the same methodology as in the *R. stolonifer* inoculation trial.

### 2.7. Quality Assessment of Treated Strawberries

The pH, total soluble solids, and total titratable acidity of strawberries subjected to the six treatments (Stw, Film, BsH, BsG, Film_BsH, and Film_BsG) were analyzed after storage at 25 °C with 40% relative humidity and at 4 °C with 70% relative humidity. Each treatment was replicated three times, adhering to a completely randomized experimental design. Strawberry juice pH was measured following the AOAC (2005) [18] guidelines. Total soluble solids were assessed using a manual refractometer (ATC-1, Atago Co., Ltd., Tokyo, Japan). For total titratable acidity, strawberry juice was titrated with 0.1 N NaOH, again following the AOAC (2005) [18] standards. All analyses were conducted in triplicate.

### 2.8. Detection of Bacillus *spp.* in Treated Samples

To evaluate the presence of *Bacillus* spp. in strawberries from the various treatments after 20 days at 25 °C or 30 days at 4 °C, we excised 0.3 cm diameter pieces from the strawberries and cultured them on PDA agar in Petri dishes (90 × 15 mm). These samples were then incubated at 25 °C for 48 h. Post incubation, *Bacillus* spp. were identified morphologically using a compound microscope at 100× magnification (Fisher S71003D), following the guidelines set by Vos et al. (2009) [19]. Each assay was conducted in triplicate.

### 2.9. Statistical Analysis

The statistical analysis of the data was performed using ANOVA, preceded by checks for data normality and homoscedasticity to ensure the validity of the test assumptions. Post hoc mean comparisons were conducted via Tukey’s test for significant differences (*p* ≤ 0.05). These analyses were executed in the R programming language (version 4.1.1) within the RStudio interface for Windows 10. Heat maps were generated using MetaboAnalyst 5.0 to visually represent the data patterns.

## 3. Results

### 3.1. Physicochemical Properties of Films with and without Bacillus subtilis

The analysis of films enriched with *Bacillus subtilis* strains GOS 01 B-67748 and HFC 103 revealed no statistically significant variations in their thickness, density, or solubility when compared to control films devoid of the bacterial inclusion. Nonetheless, a notable increase in moisture content was observed in the films containing *B. subtilis,* indicating a distinct physicochemical interaction attributed to the presence of the bacteria. This enhancement in moisture retention could have potential implications for the film’s microbial barrier properties and its overall effectiveness in food preservation applications. The detailed physicochemical characteristics of the films, both with and without *B. subtilis,* are depicted in Table 1.

### 3.2. Antifungal Effectiveness of Bacillus subtilis against Rhizopus stolonifer

#### 3.2.1. In Vitro Antifungal Efficacy

The in vitro assays to evaluate the antifungal efficacy of *Bacillus subtilis* strains against *Rhizopus stolonifer* demonstrated a notable inhibitory effect. Specifically, the application of strain GOS 01 B-67748 resulted in up to 68% inhibition of *R. stolonifer* growth after a 48 h incubation period. Similarly, strain HFC 103 exhibited a substantial inhibitory capacity, achieving a 60.6% reduction in fungal growth under identical conditions. These findings underscore the potent antifungal properties of the tested *Bacillus subtilis* strains, highlighting their potential utility in controlling *Rhizopus stolonifer* proliferation.

#### 3.2.2. Protective Efficacy on Strawberries

In a parallel assessment of protective efficacy on strawberries, control samples not subjected to *Rhizopus stolonifer* (Stw) consistently exhibited minimal damage, serving as a baseline for comparison. In contrast, strawberries coated with either the base film alone or films infused with *Bacillus subtilis* strains demonstrated a marked reduction in damage severity. This protective effect was evident across all four sampling intervals, with the bacterially infused films showing significantly less damage compared to strawberries treated with other *R. stolonifer*-exposed interventions. Notably, the strawberries coated with *Bacillus subtilis*-enriched films exhibited a level of protection that underscores the potential of these bioactive films in enhancing fruit preservation by mitigating fungal damage. Figure 2 illustrates the comparative impact of *Bacillus subtilis*-infused films on the severity of *Rhizopus stolonifer* damage in strawberries, visually reinforcing the quantitative findings and underscoring the efficacy of these bioactive coatings in protecting strawberries from fungal degradation.

### 3.3. Strawberry Preservation Assessment Using Bacillus subtilis Films

#### 3.3.1. Effect at 25 °C

In our study, the preservation efficacy of *Bacillus subtilis*-infused films on strawberries was rigorously evaluated under controlled conditions. Notably, strawberries coated with films containing the *B. subtilis* strain GOS 01 B-67748 and stored at 25 °C exhibited a marked reduction in damage severity, particularly during the later stages of the assessment period (days 15 to 27). This contrasted with strawberries treated with films incorporating the HFC103 strain, which showed increased damage from days 15 to 21, and the contrast was more pronounced when compared to other treatment groups. By the conclusion of the assessment period, all treated strawberries showed signs of advanced decomposition to varying degrees, underscoring the temporal limitations of the protective effects provided by the films. Figure 3 visually depicts the progression of damage severity in strawberries subjected to different film treatments at 25 °C, providing a clear comparison of their relative preservation efficacies.

#### 3.3.2. Effect at 4 °C

The investigation further extended to examining the effects of *Bacillus subtilis*-infused films at 4 °C, a temperature more conducive to prolonged storage. Remarkably, strawberries coated with films containing the HFC 103 strain demonstrated significantly reduced damage severity over an extended period (T4 to T9), outperforming those treated with the GOS 01 B-67748 strain (T4 to T7) and all other treatments. By the end of the evaluation period, untreated strawberries displayed the highest level of damage, highlighting the beneficial impact of the *B. subtilis*-infused films in mitigating degradation. Figure 4 provides a comparative analysis of the preservation efficacy of the different film treatments at 4 °C, illustrating the superior performance of the HFC 103 strain in enhancing strawberry longevity under refrigerated conditions.

The heatmap analysis of damage severity in strawberries subjected to various treatments at different temperatures revealed distinct patterns of clustering. This analysis identified two primary groups. The first group consisted solely of untreated strawberries (Stw), which exhibited the highest levels of damage. In contrast, the second group encompassed strawberries that had undergone some form of treatment, further divided into three subgroups based on the type of intervention applied.

Within this latter group, strawberries coated with films containing *Bacillus subtilis* strains (Film_BsG, Film_BsH) formed a distinct subgroup, showing notably reduced damage severity. Particularly, those treated with the strain GOS 01 B-67748 and stored at 25 °C demonstrated relatively lower damage severity, indicated on the heatmap in shades of blue. Meanwhile, strawberries with only the base film (Film) constituted another subgroup, and a third subgroup included strawberries directly treated with the bacterial strains (BsG, BsH).

Interestingly, strawberries stored at 4 °C exhibited lower damage severity compared to those at 25 °C, highlighting the benefits of refrigeration in fruit preservation. However, strawberries treated with the film containing the *B. subtilis* strain HFC103 and stored at 4 °C showed even lower damage severity, suggesting a particularly effective synergy between this treatment and reduced storage temperature.

Figure 5 presents the heatmap analysis of damage severity in strawberries under various treatments, providing a visual and intuitive representation of the observed patterns and underscoring the differential efficacy of treatments depending on storage temperature.

### 3.4. Quality Analysis of Strawberries Treated with Bacillus subtilis Films

#### 3.4.1. Quality Parameters at 25 °C

An exhaustive analysis of the quality parameters of strawberries treated with *Bacillus subtilis*-infused films and stored at 25 °C was conducted. The assessment focused on pH levels, total titratable acidity, and total soluble solids, which are crucial indicators of fruit quality and preservation efficacy. Despite the anticipated variations due to the bioactive nature of the films, the results indicated no statistically significant differences among the treated groups in any of the measured quality parameters. This uniformity suggests that the *Bacillus subtilis* films, while providing antifungal protection, did not adversely affect the intrinsic quality attributes of the strawberries. Figure 6 delineates the comparative analysis of these quality parameters, underscoring the consistency across the treated samples at 25 °C.

#### 3.4.2. Quality Parameters at 4 °C

Similarly, the quality assessment of strawberries treated with *Bacillus subtilis* films and stored at 4 °C revealed a consistent profile across all treated groups. The evaluation of pH, total titratable acidity, and total soluble solids yielded no significant differences, indicating that the protective efficacy of the films did not compromise the strawberries’ quality at the reduced temperature. This consistency in quality parameters at 4 °C, depicted in Figure 7, reinforces the potential of *Bacillus subtilis*-infused films as a viable strategy for extending the shelf life of strawberries without impacting their quality.

### 3.5. Presence of Bacillus *spp.* in Films Applied to Strawberries

In our comprehensive analysis of the *Bacillus subtilis*-infused films applied to strawberries, we observed a consistent presence of *Bacillus* spp. across all treated samples, irrespective of the storage conditions at 25 °C and 4 °C. The microscopic examination revealed the characteristic morphology of *Bacillus* spp.: Gram-positive, rod-shaped bacteria, often arranged in short chains or existing as individual cells. A notable feature was the presence of a central spore, a hallmark of the *Bacillus* genus, confirming the successful integration and persistence of the *B. subtilis* strains GOS 01 B-67748 and HFC103 within the film matrix over the storage period. *Bacillus subtilis* presence was confirmed solely in segments of strawberries that underwent treatment with the bacterially infused film or were directly inoculated with the bacterial culture. Due to the absence of specific molecular markers for the introduced bacteria, it was postulated that colonies manifesting morphological characteristics indicative of *Bacillus* spp. were attributable to the inoculated strains, thereby classified as *Bacillus* sp. It warrants mention that in a subset of agar plates featuring untreated strawberry segments, microbial entities characterized by mucogenic properties and coccoidal form were observed, albeit in a limited capacity, comprising approximately 10% of the samples subjected to analysis.

These findings not only underscore the resilience and stability of *Bacillus subtilis* within the film but also highlight the potential of these bioactive films to maintain viable populations of beneficial bacteria. This characteristic is crucial for the sustained release of antimicrobial compounds, which are instrumental in the protective effect observed against *Rhizopus stolonifer* and possibly other microbial threats. The presence of these bacteria, in their viable form, within the films applied to strawberries, suggests an ongoing bio-preservative action, which could be a significant factor in the extended shelf life and reduced spoilage observed in treated strawberries.

## 4. Discussion

The in vitro inhibitory capacity of *Rhizopus stolonifer* by *Bacillus subtilis* strains GOS 01 B-67748 and HFC103 demonstrated in this study is in alignment with previously reported findings on other strains, yet it showcases somewhat elevated inhibition rates. Specifically, while *B. subtilis* SM21 was reported to inhibit *R. stolonifer* by 48.9% in vitro [20], the present study observed inhibition rates of 68.8% and 60.6% for GOS 01 B-67748 and HFC 103, respectively. This indicates a variability in the antifungal activity across different *B. subtilis* strains, yet consistently highlights a potent antifungal capacity.

Further exploration into the action of *B. subtilis* SM21 applied to peach fruit via wounding at a concentration of 1 × 10^8^ CFU/mL revealed not only a reduction in the diameter and incidence of fungal lesions by 37.2% and 26.7%, respectively, but also an enhancement in the activity of antifungal enzymes β-1,3-glucanase and chitinase, coupled with an accumulation of H_2_O_2_ [20]. This suggests a complex antifungal mechanism at play. Additionally, the secretion of lipopeptides such as fengycin, surfactin, and iturin by *B. subtilis,* with iturin showing particular efficacy against *R. stolonifer* spores (MIC = 100 μg/mL) [21], further underscores the diverse and potent antifungal mechanisms employed by *B. subtilis* strains.

The comparison of the base film and *Bacillus subtilis*-integrated films revealed a similar capacity in controlling *R. stolonifer* in strawberries. However, a nuanced observation was made upon assessing the longevity of strawberries coated with these films at 25 °C and 4 °C. It became apparent that incorporating *B. subtilis* strains into the films significantly enhanced the preservation of strawberries, with each strain exhibiting optimal performance at specific temperatures. The strain GOS 01 B-67748 was particularly effective in preserving strawberries at 25 °C, while the HFC 103 strain excelled in preservation at the lower temperature of 4 °C. The temperature-dependent efficacy observed between the *B. subtilis* strains may be attributable to their origins, with GOS 01 B-67748 isolated from a temperate environment and HFC 103 from a colder region, suggesting ecological adaptations that influence their antifungal activity and interaction with the film matrix under varying temperature conditions.

The strategy of incorporating antifungal agents into films and coatings to safeguard strawberries against microbial spoilage is further supported by the existing literature. For instance, a coating comprising sodium alginate and chitosan, along with an antifungal emulsion of *Cinnamomum cassia* oil, significantly reduced fungal and yeast counts in strawberries stored at 4 °C [22,23]. Similar reductions in microbial proliferation were observed in strawberries coated with essential oils and stored at low temperatures [24], as well as in strawberries and blackberries treated with chitosan-based films and limonene-liposome edible films, respectively [25,26]. These outcomes align with the results of the current study, where *Bacillus subtilis* HFC 103 films demonstrated a reduced severity of damage from native fungi and extended shelf life at 4 °C, with a similar protective effect observed at 25 °C with *B. subtilis* GOS 01 B-67748.

The integration of microorganisms such as *Lactobacillus paracasei* TEP6 and *Wickerhamomyces anomalus* BS91 into film formulations has shown varying degrees of antifungal effectiveness [27,28], supporting the potential of microbial incorporation in edible coatings. The extension of shelf life and reduction in fungal growth in strawberries treated with a carboxymethylcellulose film containing bacteriocin from *Bacillus methylotrophicus* BM47 [29] corroborate the efficacy of microbial agents in fruit preservation.

Regarding pH stability, no significant differences were observed among treatments stored at 25 °C and 4 °C, despite temporal pH changes. This consistency is in line with previous reports, where coated strawberries did not exhibit significant pH variation over time, although control strawberries showed considerable pH increases [30]. Similar stability in total soluble solids and titratable acidity was noted in strawberries treated with various coatings and stored at 4 °C [22,23,25,26], indicating that the *Bacillus subtilis*-infused films, alongside other coating formulations, effectively maintain intrinsic quality parameters of strawberries, supporting their use in extending shelf life and preserving fruit quality under diverse storage conditions.

In summary, the findings from this comprehensive investigation not only reinforce the potent antifungal capabilities of *Bacillus subtilis* strains against *Rhizopus stolonifer* but also underscore the efficacy of *B. subtilis*-infused edible films in enhancing the shelf life and quality of strawberries across varied storage temperatures. These outcomes, set against the backdrop of the existing literature, highlight the innovative potential of microbial-based coatings in food preservation. As we move towards the conclusion, it is imperative to reflect on the broader implications of this research, particularly in the context of sustainable and safe food preservation techniques that align with current consumer preferences for natural and additive-free food products.

Our study’s limitations include the depth of film characterization and the range of antifungal mechanisms explored. Advanced microscopy could further elucidate the interaction between *Bacillus subtilis* and the film matrix, enhancing our understanding of its protective effects. Additionally, expanding our research to include various fungal pathogens would provide a broader perspective on the antifungal efficacy of *Bacillus subtilis*. Future studies will focus on these aspects to refine the application of Bacillus-infused edible films in food preservation.

## 5. Conclusions

The integration of *Bacillus subtilis* strains GOS 01 B-67748 and HFC 103 into a guar gum, candelilla wax, and glycerol film notably enhanced the film’s moisture retention capabilities without affecting its thickness, density, or solubility. This modification suggests that the presence of *B. subtilis* influences the film’s moisture interactions, potentially contributing to its protective qualities.

Both the base film and the *B. subtilis*-infused films effectively mitigated *Rhizopus stolonifer* contamination in strawberries. Notably, films containing *B. subtilis* extended the freshness and shelf life of strawberries beyond that achieved by the base film alone. Among the strains, GOS 01 B-67748 exhibited superior protective performance at 25 °C, whereas HFC103 was more effective at 4 °C, indicating a temperature-dependent efficacy that might be linked to the strains’ ecological origins and adaptations.

Quality parameters such as pH, total soluble solids, and total titratable acidity showed no significant differences between strawberries treated with *B. subtilis*-infused films and those subjected to other treatments, regardless of storage at 25 °C or 4 °C. This consistency underscores the films’ ability to preserve strawberry quality without altering fundamental quality attributes.

Furthermore, *Bacillus* spp. were detected on all strawberries treated with *B. subtilis* films or directly with the bacteria, confirming the successful incorporation and persistence of these beneficial microbes on the fruit surface.

The findings of this study underscore the considerable potential of *Bacillus subtilis* strains in film formulations to control *R. stolonifer* and enhance the shelf life of strawberries. Future research should explore the efficacy of *B. subtilis*-infused films across a wider range of storage temperatures to fully understand the scope of their protective effects on fruit preservation.

## Figures and Tables

**Figure 1 foods-13-00980-f001:**
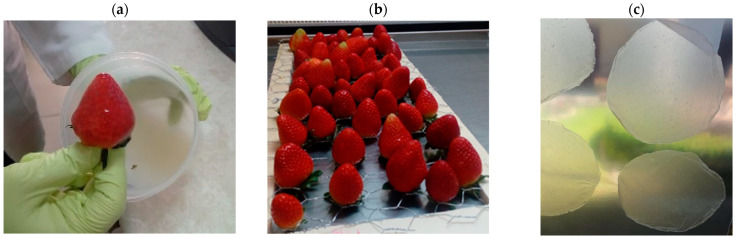
Edible films with *Bacillus subtilis*. (**a**) Application of edible films containing *Bacillus subtilis* on strawberries; (**b**) example of strawberries coated with edible films incorporating *Bacillus subtilis*; (**c**) preformed films containing Bacillus subtilis for characterization purposes.

**Figure 2 foods-13-00980-f002:**
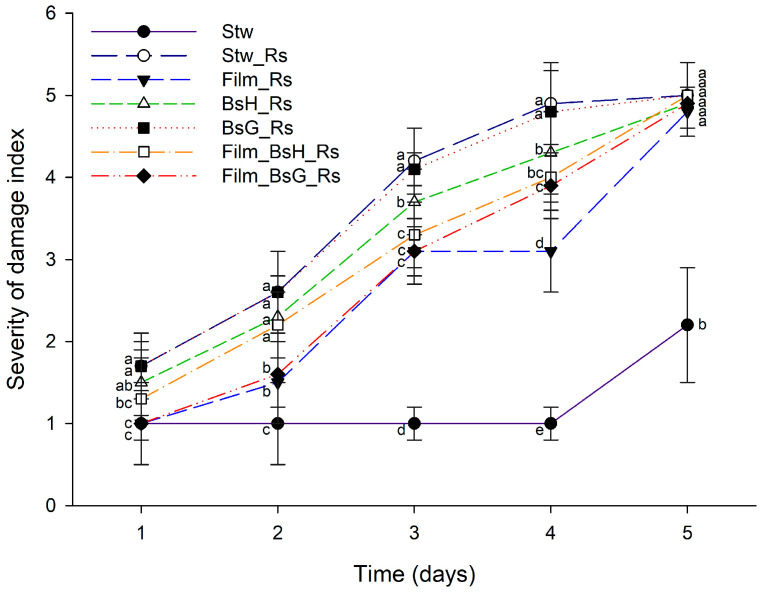
Impact of *Bacillus subtilis*-infused films on *Rhizopus stolonifer* damage in strawberries over a 5-day period. The graph displays daily mean values along with their standard deviations. Different superscript letters indicate statistically significant differences between treatments for each day, determined by ANOVA and Tukey’s post hoc analysis (*p* ≤ 0.05).

**Figure 3 foods-13-00980-f003:**
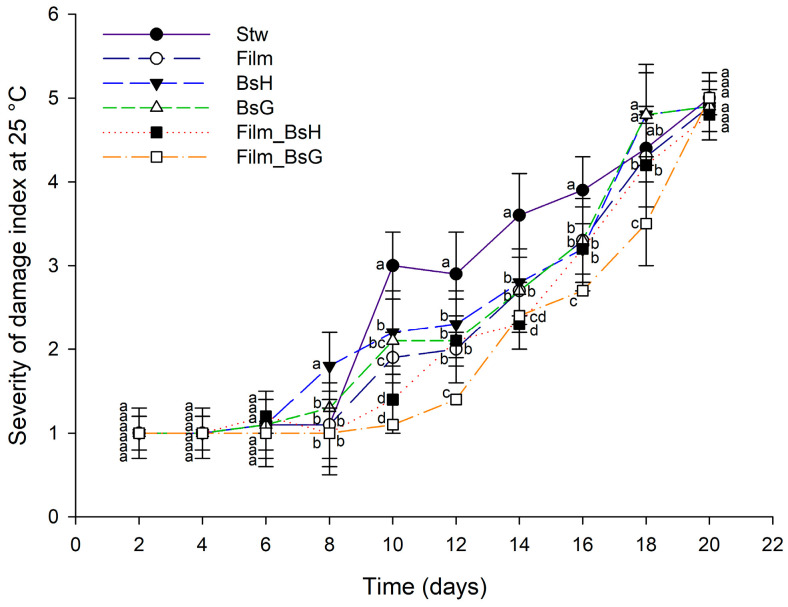
Damage severity in strawberries with various film treatments at 25 °C over a 20-day period. The graph displays daily mean values along with their standard deviations. Different superscript letters indicate statistically significant differences between treatments for each day, determined by ANOVA and Tukey’s post hoc analysis (*p* ≤ 0.05).

**Figure 4 foods-13-00980-f004:**
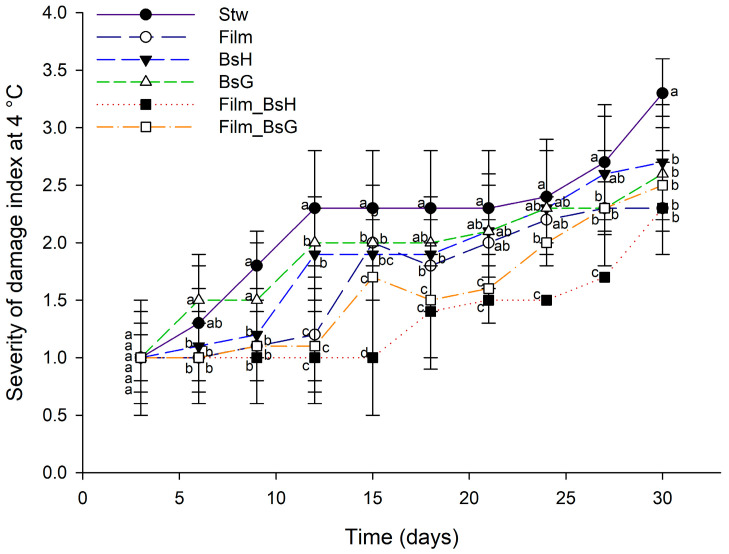
Evaluation of strawberry preservation at 4 °C using *Bacillus subtilis*-infused films over a 30-day period. The graph displays daily mean values along with their standard deviations. Different superscript letters indicate statistically significant differences between treatments for each day, determined by ANOVA and Tukey’s post hoc analysis (*p* ≤ 0.05).

**Figure 5 foods-13-00980-f005:**
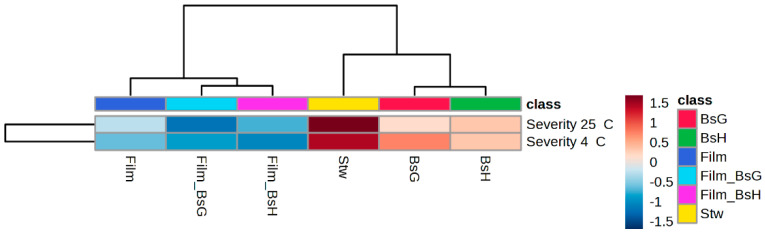
Heat map analysis of damage severity in strawberries under various treatments. Rows: severity of damage. Columns: strawberry treatments. The correlation matrix (Euclidean) presents a color gradient where the blue colors indicate a positive correlation and the red colors a negative correlation. The color groupings are shown at the top of the figure, where the strawberries with films containing *B. subtilis* form one subgroup, the strawberries with the base film another, and the strawberries with the bacteria another.

**Figure 6 foods-13-00980-f006:**
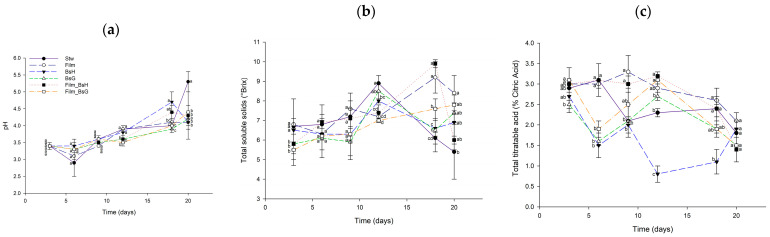
Quality parameters of strawberries treated with *Bacillus subtilis* films at 25 °C over a 20-day period. The graph displays daily mean values along with their standard deviations. Different superscript letters indicate statistically significant differences between treatments for each day, determined by ANOVA and Tukey’s post hoc analysis (*p* ≤ 0.05). The evaluated parameters include (**a**) pH, (**b**) total soluble solids (°Brix), and (**c**) total titratable acidity (% citric acid).

**Figure 7 foods-13-00980-f007:**
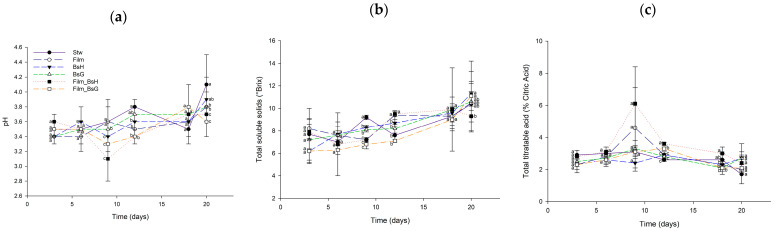
Quality parameters of strawberries treated with *Bacillus subtilis* films at 4 °C over a 20-day period. The graph displays daily mean values along with their standard deviations. Different superscript letters indicate statistically significant differences between treatments for each day, determined by ANOVA and Tukey’s post hoc analysis (*p* ≤ 0.05). The evaluated parameters include (**a**) pH, (**b**) total soluble solids (°Brix), and (**c**) total titratable acidity (% citric acid).

**Table 1 foods-13-00980-t001:** Comparative physicochemical analysis of edible films with and without *Bacillus subtilis* inclusion.

Treatment	Thickness (mm)	Density (g/cm^3^)	Solubility (%)	Moisture (%)
Film	0.3 ± 0.02 ^a^	0.02 ± 0.03 ^a^	24.23 ± 5.86 ^a^	20.6 ± 0.64 ^b^
Film_BsH	0.4 ± 0.03 ^a^	0.02 ± 0.00 ^a^	24.76 ± 3.57 ^a^	27.8 ± 0.56 ^a^
Film_BsG	0.6 ± 0.02 ^a^	0.03 ± 0.03 ^a^	22.27 ± 4.08 ^a^	28.2 ± 2.43 ^a^

Means ± standard deviations are presented. Different superscript letters represent a significant difference between film formulations, as determined by ANOVA followed by Tukey’s test (*p* ≤ 0.05).

## Data Availability

The original contributions presented in the study are included in the article, further inquiries can be directed to the corresponding authors.

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
