# Peer review of "Bacillus subtilis Edible Films for Strawberry Preservation: Antifungal Efficacy and Quality at Varied Temperatures"

_foods, 2024, doi:10.3390/foods13070980_

Round 1

Reviewer 1 Report

Comments and Suggestions for Authors

I find this work very interesting and very topical in terms of strategies for the use of edible films in food. I found it very well presented and easy to read, with the right methodology However I have some doubts and some aspects that authors should clarify:

1. First of all, about the safety of these B. subtilis strains. The authors indicate in the introduction (line 62) that this bacterial species is recognised as GRAS, however, the safety must be assessed for the strains to be used and the recognition must be given for the strains to be used in the feed. I do not know whether in this respect the strains that have been evaluated in this work have been tested to verify, among other things, the absence of resistance genes mentioned in the introduction.

This seems to me to be a key aspect when it comes to evaluating other aspects such as consumer acceptability, sensory analysis, etc... in addition, it is necessary to think about the impact that the ingestion of these bacteria could have on the organism.

2. Secondly, I would like to ask about the appearance of the films. It might be interesting to include some pictures, both of the films in the Petri dishes and on the strawberries and how this aspect varies during storage. I would also like to ask about the morphology of the film and if the authors have used any technique such as scanning electron microscopy for this.

3. Finally, I would like to ask about the cell viability of the B. subtilis strains over time. The authors indicate that they have carried out microscopic determinations to confirm the integration and persistence of the bacteria in the film matrix but it is not clear to me how they have been able to determine this aspect because of the following:

- the authors indicate that they use PDA (Potato Dextrose Agar) to grow the strains, but this medium is not specific to Bacillus spp. but is used for the cultivation of fungi and yeasts.....

- As it does not use a specific medium for Bacillus, in this case neither for bacteria, how do the authors know when assessing cell viability that these colonies are Bacillus? Especially when assessing cell viability, as the strawberries will have their own microbiota, and what grows from these cultures may not be Bacillus spp.

In my opinion, the authors should clarify all these aspects that generate doubts when cultivating and evaluating the integration and viability of Bacillus spp. strains.

Reviewer 2 Report

Comments and Suggestions for Authors

This study integrated Bacillus into edible films, measured its physicochemical properties, and revealed its potential in strawberry preservation. The manuscript cannot be accepted in its current form, which needs to be reconsidered with major revision. The following are my suggestions for improving the manuscript.

1. The characterization methods for thin films in this study are limited which only include the most basic physical and chemical properties. The authors need to provide photos of the film and its application on strawberries. Deeper characterization means are necessary, such as SEM, etc.

2. Although the antifungal mechanism of Bacillus has been extensively studied, this article did not cover the direct antifungal mechanism. There is a lack of determination of the number of active Bacillus species in the film and of the key substances responsible for its direct antifungal activity (e.g. specific volatile components). Antifungal activity was determined only against Rhizopus stolonifer and other important postharvest pathogens were not included.

3. The rationality of some data is debatable.

Line 88-89, what means "high-quality materials"? Does it mean "high molecular weight" or the chemical purity?

Line 89-96, the authors should provide specific information about these reagents, including manufacturer and specific purity, etc. The rationale for your choice should appear in the introduction, not the Materials Methods section.

Line 98, since B. subtilis is a bacteria, why did not the author culture it on LB media?

Line 100-101, what was the pH value of the suspension?

Line 109, the 20% is the proportion to the whole solution? I recommend the author provide a table of film solution compositions.

Line 121, "25°C ± 5" should be revised to "25±5°C". Same at line 122.

Line 194, what was the relative humidity?

Line 225, the authors should provide the picture of the films.

Figure 2, is it necessary to store strawberries at 25°C for 20 days? On days 18 and 20, the upper limit of the error bars exceeds 5, which indicates that the damage has exceeded 100% (at line 178). Unless the authors can provide photos of the fruit, the rationality of this part of the data is debatable.

Figures 3, 5, and 6, the figures need to be re-drawn. The data points cannot correspond to the coordinate axes.

Line 339-347, there is no data for section 3.5.

Round 2

Reviewer 2 Report

Comments and Suggestions for Authors

The authors improved the manuscript.

Comments for improving the manuscript:

Line 189-190, since damage severity is the most critical evaluation the article, the authors should provide the typical pictures of different scales, also the symptoms of the damage (either physiological, microbial, mechanical damage or so).

Figure 6B-C, 7B-C, there are no units in those figures.

Figure 4, 6, 7, please address the sample date in the caption.
